# Oral Mucositis in Adult Cancer Patients Undergoing Chemotherapy: Six-Month On-Treatment Follow-Up

**DOI:** 10.3390/jcm13195723

**Published:** 2024-09-25

**Authors:** Adriana Padure, Raluca Horhat, Ioana Cristina Talpos-Niculescu, Roxana Scheusan, Mirella D. Anghel, Laura-Cristina Rusu, Diana Lungeanu

**Affiliations:** 1Multidisciplinary Center for Research, Evaluation, Diagnosis and Therapies in Oral Medicine, “Victor Babes” University of Medicine and Pharmacy, 300041 Timisoara, Romania; adriana.padure@umft.ro (A.P.); laura.rusu@umft.ro (L.-C.R.); 2Clinic of Oro-Dental Diagnosis and Ergonomics, “Victor Babes” University of Medicine and Pharmacy, 300041 Timisoara, Romania; ioana.talpos-niculescu@umft.ro (I.C.T.-N.); ergonomie@umft.ro (M.D.A.); 3Center for Modeling Biological Systems and Data Analysis, “Victor Babes” University of Medicine and Pharmacy, 300041 Timisoara, Romania; dlungeanu@umft.ro; 4Department of Functional Sciences, “Victor Babes” University of Medicine and Pharmacy, 300041 Timisoara, Romania; 5Oncocenter Oncologie Clinica S.R.L, 300166 Timisoara, Romania; roxana.scheusan@oncocenter.ro; 6Clinic of Oral Pathology, “Victor Babes” University of Medicine and Pharmacy, 300041 Timisoara, Romania

**Keywords:** oral hygiene, cancer, oral mucositis, cumulative incidence, lesion-free survival, cancer patient education, dental care, oral health, chemotherapy, basic oral care

## Abstract

**Objectives**. Oral mucositis (OM) is a common adverse reaction associated with chemotherapy. We conducted a six-month longitudinal study to estimate the cumulative incidence of OM during the first six months of chemotherapy in adult patients with cancer other than head and neck cancer. Secondary objectives were as follows: (a) to scrutinize the oral health status of these patients and its evolution during chemotherapy, as assessed by oral health indices; (b) to estimate adherence to prescribed oral hygiene protocol during chemotherapy; and (c) to analyze ulceration-free survival in these patients. **Methods**. Sixty-four patients participated. Dental health and oral hygiene were assessed at baseline and at the end. Every month, blood tests were performed and oral lesions were recorded. This study was observational, with the only intervention being instruction in the hygiene protocol. The cumulative incidence of OM was estimated with the patient as the unit of analysis. A repeated measures ANOVA was applied to analyze the monthly blood test results. Ulceration-free survival analysis was conducted with adherence to the oral hygiene protocol as a grouping factor, followed by Cox proportional-hazards regression. **Results**. The six-month cumulative incidence rate was 43.75%, 95%CI (31.58–56.67%) for OM grade 2 or higher. The hazard ratio of ulceration associated with adherence to the hygiene protocol was 0.154, 95%CI (0.049–0.483), adjusted for age, sex, baseline hygiene index, and class of treatment. **Conclusions.** Compliance with hygiene recommendations would decrease the OM risk by more than six times, compared to non-compliance.

## 1. Introduction

Cancer is the second most frequent cause of death, causing millions of deaths each year [1]. Chemotherapy has an important role in its treatment, in spite of being associated with substantially distressing side effects, such as oral mucositis (OM) [2,3,4,5,6,7,8,9,10,11,12,13]. Severe OM interferes with oral intake, sometimes leading to parenteral nutrition and an increased need for systemic analgesics. Moreover, OM can lead to bacteremia or sepsis, thus prolonging the hospital stay and even altering or interrupting the chemotherapy itself [2]. The overall increase in treatment costs can be up to 6000 USD [14]. At the same time, OM makes oral hygiene more difficult, increasing the presence of food debris in the oral cavity; when associated with chemotherapy-induced xerostomia, this leads to a higher risk for oral cavities [15].

During chemotherapy, DNA damage occurs and reactive oxygen species are released, causing basal and suprabasal mucosal cell death [3,16], which triggers the primary damage response. Nuclear factor-kB (NF-κB) modulates the transcription of more than 200 genes associated with pro-inflammatory cytokines, such as IL-6, IL-1β, and TNFα, thereby inducing early connective and epithelial tissue damage and inhibiting tissue oxygenation that further contributes to basal epithelial cell death [16,17,18]. Positive feedback loop mechanisms lead to the amplification of the injury signal. TNF-α activates the mitogen-activated protein kinase (MAPK) and sphingomyelinase, increasing apoptosis and supporting NF-kB activity [3]. At this stage, although tissue damage is present, patients usually complain little, with the next stage of the inflammatory process being more clinically significant [6,19]. The presence of ulceration is characterized by pain and the possibility of infection by symbiotic microorganisms, and a cell-mediated inflammatory response will further increase tissue damage. Discontinuation of chemotherapy is associated with the onset of the healing process and OMs usually self-resolve [3,20].

The incidence and severity of OM depend on many factors related to the treatment (such as type, dose, the timing of chemotherapy sessions, or its association with radiotherapy) or to the patient (such as age, sex, body mass index, salivary function, oral health status, and pre-existing mucosal trauma) [18]. Genetic determinants and oral microflora also play a role in the pathogeny of mucositis [6,12,21]. Oral mucosa regeneration is influenced by changes in the DNA methylation profile. Recent studies have found hypomethylation in the promoter of the pro-inflammatory cytokine TNF-α gene [22], as well as DNMT1 profile, to be associated with hypomethylation of pro-inflammatory genes [23]. Moreover, rs1544410 and rs2228570 polymorphisms in the vitamin D receptor gene were found in patients with severe OM [24]. Resident oral bacteria prevent colonization by exogenous microorganisms, preventing local and systemic infection. Although there was no identified pattern of oral microflora changes in chemotherapy patients, the most frequent Gram-negative isolated species are from the *Enterobacteriaceae* family, *Pseudomonas* species, and *Escherichia coli*. The most frequent Gram-positive are *Staphylococcus* species and *Streptococcus* species [25]. *Porphyromonas gingivalis* has been identified in patients with ulcerative lesions of OM, a bacterium associated with periodontitis [26]. Viruses such as herpes-simplex were also found to play a role in OM development [27].

Inflammation markers such as the neutrophil-to-lymphocyte ratio (NLR) have been shown to be predictive factors for radiotherapy-induced OM in head and neck cancer patients [11,28]. Tissue damage in OM results from both apoptosis and tissue necrosis through an inflammatory process [6]. The systemic inflammatory response is associated with alterations of the white blood cell counts, such as neutrophilia and relative lymphopenia, leading to changes in the NLR [11,29]. Neutrophils increase the release of vascular endothelial growth factors (VEGF), which stimulates the release of inflammatory cytokines such as IL-1, IL-6, and TNFα, also implicated in the pathogenesis of OM [14,28,30]. Patients with increased levels of pro-inflammatory cytokines and peritumoral macrophage infiltration also exhibit elevated NLR [31,32].

OM is a common clinical manifestation (up to a rate of 85%) in patients with head and neck cancer [5,33,34,35], making it difficult to determine the adequate doses of chemotherapy and radiotherapy targeting these very areas, namely the head and neck. Moreover, radiotherapy has a central role in head and neck cancers (especially in locally advanced lesions), and the tumor lesions are associated with mucosal ulcerations or oral infections, involving local anatomical structures [34]. On the other hand, OM also occurs in patients with different cancer locations and undergoing treatment targeting other areas of the body (a rate of 30–40%) [5]. Pondering the dissimilarities and due to the substantially higher prevalence of oral mucositis in head and neck cancer cases, the clinical particularities, and the therapeutic approach usually involving radiotherapy in head and neck cancer, our analysis excluded patients with head and neck cancer ab initio.

Given the possible complications, medical burden, and implications of OM on patient prognosis, several preventive measures have been investigated: basic oral care (BOC), use of anti-inflammatory agents, photobiomodulation therapy, cryotherapy, administration of antimicrobials, analgesics, growth factors, and cytokines, or natural compounds [2,13,20,36]. The method that has received the highest level of evidence according to the Multinational Association of Supportive Care in Cancer/International Society for Oral Oncology (MASCC/ISOO) guidelines [2,37] is good clinical BOC practice. Dental assessment is necessary before the start of chemotherapy, followed by professional oral care during treatment. Combined oral care protocols with multiple agents are recommended and patient education is encouraged. Oral hygiene protocols can be associated with saline, sodium bicarbonate, or antiseptic mouthwashes [2,37].

The Romanian population has particular characteristics in terms of oral health, knowledge of oral care, and cultural backgrounds, with poor statistics regarding oral health and levels of oral care literacy in all age groups [38,39,40,41,42,43]. This context makes cancer patients more likely to suffer from OM, and little is known about the incidence of these oral health problems among cancer patients in Eastern Europe.

The primary objective of this study was to estimate the cumulative incidence of OM during the first six months of chemotherapy in adult patients with cancer other than head and neck cancer.

The secondary objectives were as follows: (a) to scrutinize the oral health status of these patients and its change during the chemotherapy, as assessed by indices of oral health; (b) to estimate the adherence to prescribed oral hygiene protocol during the chemotherapy; and (c) to analyze ulceration-free survival in these patients.

## 2. Materials and Methods

### 2.1. Study Design

We conducted a prospective, longitudinal study. The initial study group consisted of 71 consecutive patients diagnosed with cancer (other than head and neck cancer) and admitted for chemotherapy at Oncocenter, a private oncology clinic in the Western part of Romania, between 1 November 2022 and 30 June 2023. The follow-up period was six months. Figure 1 shows the study flowchart.

The study protocol was reviewed and approved by the Ethics Committee of “Victor Babes” University of Medicine and Pharmacy in Timisoara, Romania. All participants in the study gave their written informed consent.

Inclusion criteria were as follows: (1) age at least 18 years; (2) presence of a malignant solid tumor; (3) absence of chemo-/radiotherapy in the previous six months; (4) patient following a chemotherapy regimen; (5) absence of infection; and (6) normal C-reactive protein (CRP) values (i.e., less than 5 mg/L). The last criterion sought to avoid confounding from pre-existing inflammation, and no further distinction was made between possible sources of baseline inflammation (such as CRP elevation due to cancer versus other causes).

Patients were excluded if at least one of the following was present: (1) pre-existing oral mucosal lesions; (2) concomitant radiotherapy to the head and neck region; (3) oral cancer or premalignant disease; (4) diseases associated with oral mucosal lesions; and (5) corticosteroid treatment one month before the chemotherapy started. Drug rash and erythema, as well as Stevens–Johnson syndrome, were carefully considered for possible exclusion on a case-by-case basis.

The baseline assessment was performed before the start of chemotherapy, followed by monthly assessments during the six-month follow-up. Seventy-one consecutive patients were enrolled. One of them was excluded because he was on an immunotherapy regimen. During the follow-up period, two participants dropped out and four died, thus being excluded from the final analysis.

### 2.2. Baseline and Six-Month Assessments

Oral examination was performed on seated patients under artificial lighting by using a dental consultation kit. The patients were assessed for abnormal changes in the oral mucosa. The presence of the dental units, dental caries, dental fillings, and fixed or mobile prosthetic restorations was recorded. Dental health was assessed based on the Decayed, Missing, and Filled Permanent Teeth index (DMFT index), as recommended by the World Health Organization (WHO) [44,45]. Its values range from 0 to 32 and describe the amount (i.e., the prevalence) of dental caries in an individual. The maximum value for DMFT is 28 for 28 teeth and 32 for 32 teeth. Of the 71 patients initially included in the study, 9 were edentulous and were therefore excluded from the analyses including or requiring oral indices. Dental hygiene was quantified according to the dental hygiene index [46], which is detailed in Table 1, to assess the presence of plaque in different areas of the teeth as an indirect indicator of the hygiene habits and the effectiveness of plaque and tartar removal. The dental hygiene index per tooth corresponds to the arithmetic mean of the scores of different surfaces. The dental hygiene index per individual corresponds to the arithmetic mean of the plaque indices of the teeth evaluated. A dental file was created for each patient upon the visual examination. This assessment was repeated at the end of the follow-up period.

### 2.3. Intervention

All patients were instructed in the hygiene protocol, which consisted of (1) use of a soft-bristled toothbrush; (2) rinsing the mouth with a saline and bicarbonate solution (5 to 6 times per day, with 1 tablespoon of each salt in half a liter of water); (3) monthly brush change; (4) brushing teeth and gums 3 times a day with vertical and/or circular movements; (5) use of fluoride mint-free toothpaste; and (6) at least one flossing per day. Written instructions were provided to each patient and instruction was reinforced at subsequent assessments. We considered a patient to be compliant with the hygiene protocol if (s)he complied with it at least five days a week and twice per day throughout the six-month follow-up.

### 2.4. Monthly Assessments

Laboratory investigations were performed monthly for each patient during the follow-up period. Chemotherapy class was also recorded at each assessment. Photographs were taken at enrollment and at monthly check-ups and were sent to a second oral medicine specialist for independent confirmation.

All study participants received routine care during their hospitalization and were assessed before each treatment cycle using the oncology clinic’s questionnaire regarding general conditions and adverse effects arising from the treatment. Five to seven days after each chemotherapy session, they were contacted by the dentist. An interview regarding the compliance with the hygiene protocol was carried out each time. Newly developed lesions of the oral mucosa were reported, assessed, and photographed. The severity of the oral mucositis was graded according to the WHO scale for oral mucositis [4,5]. Patients developing various degrees of mucositis were prescribed topical analgesics, antiseptics, and corticosteroids.

### 2.5. Data Analysis

Descriptive statistics included the following: the observed frequency counts and percentages for categorical variables; the sample’s mean and standard deviation (SD) for numerical variables, irrespective of their distribution. The normality of numerical variables was tested with Kolmogorov–Smirnov statistical test (all numerical variables had slightly non-normal distributions).

Univariate non-parametric statistical tests were applied to compare the distribution of numerical data across two or multiple groups, as appropriate (either Mann–Whitney U or Kruskal–Wallis tests, respectively). The chi-square statistical test (either asymptotic or Monte–Carlo simulation with 10,000 samples) was applied to check the statistical significance of the association between the categorical variables.

Repeated measures analysis of variance (ANOVA) was applied to analyze the blood tests over the six-month follow-up and assess the statistical significance of observed trend differences between the classes of treatment. When the assumption of sphericity was not met (as was the case in this data set), the Greenhouse-Geisser correction was applied.

Estimation of the cumulative incidence rate of oral mucosal lesions as a measure of risk was performed with the patient as the unit of analysis. All patients were followed from the beginning of chemotherapy over the next six months. For the cumulative incidence rate estimation and survival analysis, cases with new lesions were counted, irrespective of duration for each patient. Incidence was estimated for (a) any type of lesion (i.e., mucositis any grade) and (b) painful inflammation and ulceration (i.e., mucositis grade 2 or higher). Survival analysis was conducted with mucositis grade 2 as the event sought in the analysis, and adherence to the protocol of oral hygiene as the grouping factor. Kaplan–Meier curves were employed as estimators of survival, and the log-rank test was applied to compare the incidence rates between the two self-selected groups of oral hygiene (i.e., adherence versus non-adherence). The Cox proportional-hazards model was applied for the estimation of the hazard ratio of painful inflammation and ulceration. Multicollinearity was tested based on the variance inflation factor (VIF). To select the best fitting model (namely, the trade-off between the goodness of fit and the simplicity of the model), the Akaike information criterion (AIC) was used.

The statistical analysis was conducted at a 95% level of confidence and a 5% level of statistical significance. All reported probability values were two-tailed. Data analysis was performed with the statistical software IBM SPSS v. 20 and R v. 4.3.1 packages (including “survival” v. 3.5-5 and “rms” v.6.6-0).

## 3. Results

### 3.1. Characteristics of Cancer Patients Enrolled

The general characteristics of the patients are shown in Table 2. The mucosal lesions were also classified according to the WHO classification and the grades of the worst experience of each patient are presented in Table 3.

The patients underwent seven classes of treatment; each received a numerical code shown in Table 4. Classes were treated as categorical variables throughout the analysis (numbers had symbolic rather than rank-ordering meaning). During the six-month follow-up, a patient could undergo different treatments.

Laboratory test results during the six-month follow-up are summarized in Table 5 and Figure 2.

Repeated measures ANOVA was applied to analyze the blood test values during the follow-up and assess the statistical significance of observed trend differences between the classes of treatment (Table 6). The highly significant effect of the treatment is apparent both in the numerical results in Table 6 and in the line graphs depicted in Figure 2. The time effect (namely, the month) did not reach statistical significance in all blood investigations. This lack of apparent trend can be observed in Figure 2 as well.

### 3.2. Incidence of Oral Mucositis

Table 7 synthesizes the estimation of the six-month cumulative incidence rate of mucositis.

### 3.3. Assessment of Six-Month Change of Oral Health

The dental hygiene and DMFT indices at baseline, at six months, and the corresponding differences are summarized in Table 8.

The DFMT indices were significantly different in compliant vs. non-compliant patients (both at baseline and at six-month check-ups), but they did not significantly deteriorate over six months. On the other hand, the increased initial value of the dental hygiene index and non-compliance with the oral hygiene protocol was associated with a significantly greater increase in the score, suggesting a relation with the OM incidence and evolution.

Moderate or severe OM lesions show a significantly higher proportion in the non-compliant group (63.5%) compared to the compliant patients (33.3%).

### 3.4. Analysis of Ulceration-Free Survival

This analysis included 55 non-edentulous patients. Table 9 synthesizes the ulceration-free survival analysis taking into account the self-reported compliance with the recommended hygiene protocol, and Figure 3 shows the Kaplan–Meier curves.

The mean time for ulceration-free survival over the first six months of chemotherapy is 5 months for patients who comply with the hygiene protocol, compared to 4 months for those who do not comply.

While the difference in mean time is about one month, there is high variability between patients. Cox regression analysis would allow quantifying the actual contribution of hygiene protocol to decreasing the risk. The results are summarized in Table 10.

The VIF values ranged from 1 to 2.286, meaning there was no collinearity of the independent variables in the regression model.

The R code for this analysis and the results in full are provided as Appendix A.

A low and statistically significant hazard ratio is associated with the hygiene protocol compliance, namely, 0.154, 95%CI (0.049–0.483), adjusted for age, sex, baseline hygiene index, and class of treatment. Adherence to the prescribed hygiene protocol leads to an important and highly significant reduction in the risk of developing oral ulceration compared to non-adherence in patients of the same age, sex, baseline hygiene index, and undergoing the same class of treatment.

Failure to adhere to strict oral hygiene recommendations would increase the risk of OM grade 2 or higher by more than six times (6.494 = 1/0.154).

## 4. Discussion

OM is a common adverse reaction associated with chemotherapy, with variable incidence and degree of severity, correlated with the classes of medication and protocols used. While the complication occurs in approximately 40% of the patients receiving standard dosing regimens, it was reported to reach percentages of 100% in patients receiving high-dose chemotherapy or in those who have undergone stem cell or bone marrow transplantation [2,12,47,48,49,50].

Our applied research aimed to estimate the cumulative incidence of OM during the first six months of chemotherapy in adult patients with cancer other than head and neck cancer, and to assess changes in oral health during chemotherapy. Our study was observational, the only intervention being instruction in the hygiene protocol. We found that adherence to the oral hygiene protocol was an independent protective factor against oral ulceration (OM grade 2 or higher), controlling for age, sex, baseline hygiene index, and chemotherapy treatment class.

The overall six-month cumulative incidence of chemotherapy-induced OM was 53.13%, 95%CI (40.33–65.55%) in our study, with the cumulative incidence of oral ulceration being somewhat lower, namely, 43.75%, 95%CI (31.58–56.67%). In a recent meta-analysis with results from 82 articles, a prevalence of 42.9%, 95%CI (32.8–53%) was reported in patients treated with chemotherapy alone [51]. Incidence and prevalence are distinct epidemiological measures, but they are related: prevalence is influenced by both the rate at which new cases are occurring (i.e., incidence) and by the average duration of the condition. Therefore, in this context of a six-month longitudinal follow-up of patients undergoing chemotherapy who had no pre-existing oral mucosal lesions (but who were undergoing symptomatic oral treatment for new lesions), we can conclude that the incidence values we found are in agreement with the incidence and prevalence previously reported. Other recent studies, with observational cross-sectional design [52] or quasi-experimental design [53], reported similar proportions of cancer patients experiencing OM. We believe that our investigation had a balanced approach between ethical principles (an educational intervention and oral medical care were offered to all participants) and a longitudinal design that allowed estimation of both cumulative incidence of chemotherapy-induced OM and ulceration-free survival time.

When analyzing the OM incidence, we must take into account the classes of chemotherapy used for treatment, as some of them (e.g., 5-fluorouracil, docetaxel, alkylating agents) are known to be more aggressive on the oral mucosa [3,9,28,29,30,31,32,50,54]. In our investigation of blood test results during the six-month chemotherapy follow-up, repeated measures ANOVA also revealed significant effects of treatment, even when no significant time trend was apparent. This latter finding in our investigation could be a consequence of changing treatment when the patient’s response was not favorable.

Clinical studies and guidelines [2,10,37,55] highlight the importance of oral hygiene in the prevention of OM, such as commercial solutions containing chlorhexidine and benzydamide with analgesic, anti-inflammatory, and antimicrobial properties, but sometimes discontinued due to the associated stinging sensation, taste alteration, and tooth staining [19]; chlorhexidine, the use of which was restricted by the MASCC/ISOO guidelines in patients with head and neck cancer undergoing radiotherapy, namely, level of evidence III (LoE III) [2,37]; saline and sodium bicarbonate mouth rinses to increase oral clearance, contributing to better oral hygiene and patient comfort [2,37]. OM is the result of complex interactions occurring in epithelial and connective tissue compartments, involving the activity of pro-inflammatory cytokines that cause apoptosis and tissue necrosis [16]. The severity of the lesions is increased by alterations of the ecological balance in the oral cavity [6] and salivary hypo-function [7], with resident oral bacteria preventing colonization by exogenous organisms, thus creating competition for ligands and receptors available for attachment in endogenous substrates like nutrients and cofactors. The products of their metabolism modify the pH and redox potential, so that the microenvironment becomes unsuitable for colonization by pathogens. Hydrogen peroxide produced by *Streptococcus oralis*, bacteriocin from the Gram-positive bacteria, and acidic end-products of bacterial metabolism act as inhibitory substances for exogenous bacteria in the mouth [6]. Therefore, rinsing solutions should respect the resident bacteria in the oral cavity, thus preventing infection. Studies analyzing the use of antibiotic lozenge containing polymyxin E, tobramycin, and amphotericin B (PTA) have shown no impact on the OM incidence and severity [8,56,57]. Chemotherapy-induced hypo-salivation is associated with a deficiency of antimicrobial factors—such as lactoferrin, lactoperoxidase, lysozyme, and defensins—as well as mucosal protectors, like sIgA or epidermal growth factor. On the other hand, different systemic chemotherapeutic agents are secreted directly into saliva. Studies have shown an increase in salivary secretion for methotrexate, doxorubicin, 5-flourouracyl, etoposide, melphalan, carboplatin, and taxol [8,58]. In these patients, frequent use of rinse solutions would dilute the oral concentrations of chemotherapeutic agents, minimizing their local effect on the mucosa and reducing the discomfort caused by hypo-salivation.

We applied the clinical practice guidelines of the MASCC/ISOO study group [2,37]. A recent systematic review and network meta-analysis regarding the efficacy of topical agents in oral mucositis prevention [36] concluded that topical sucralfate would be the most effective in OM prevention, although the MASCC/ISOO guidelines had ranked sucralfate as LoE II [2,37]. On the other hand, only 60% of patients in our study complied with the recommended hygiene protocol. They showed a significantly lower incidence of OM grade 2 or higher, and a significantly less deterioration in the hygiene index at six months compared with the non-compliant group, confirming previous findings on the beneficial use of saline and bicarbonate solutions for the prevention of OM [2,10,55,59]. Comparing the evolution of the values of dental hygiene indices between compliant and non-compliant patients, we found a significant difference in their overall oral hygiene. Moreover, there was a significant difference between the scores at baseline and after the six-month follow-up, which is probably associated with their BOC routine and lifestyle habits. In addition, it should not be overlooked that prophylactic measures were mainly carried out outside the clinic. Noncompliance occurred particularly in elderly patients with low incomes, little family support, or with advanced malignancy and poor general condition.

Studies analyzing the side effects of chemotherapy tend to focus mainly on OM. The few reports regarding oral health in this category of patients have been retrospective or cross-sectional studies [49,60,61], or have been performed on a pediatric population [47,48]. When comparing six-month DMFT indices in our sample with other studies, the values were somewhat higher [49]. This could be partly explained by the peculiarities of the dental health system in Eastern Europe (Romania in particular), where dental services are provided by dentists working in private practices and clinics. Patients’ expenses are only partially reimbursed by the National Health Insurance Fund, with the budget being insufficient for patients‘ needs [62]. Moreover, services are mainly focused on treatment, while prevention benefits are often neglected [42]. The Romanian population is recognized for its poor oral health and its determinant socio-behavioral risk factors [38,39,40]. Nevertheless, in our study, adherence to the hygiene protocol has been shown to significantly reduce the risk of oral ulceration by more than six times in cancer patients of the same age, sex, and baseline hygiene index irrespective of their treatment. To achieve better overall oral health and a subsequent lower rate of chemotherapy-induced oral complications, sustained health education and prevention are necessary [63].

### Study Limitations

The cumulative incidence of mucositis estimated in this study has limited external validity due to the inherent limitations of single-center research. On the other hand, it is important to have such estimates for populations in diverse geographic areas, with their cultural, socioeconomic conditions, and public health backgrounds. The prospective design brings validity and reliability to this specific estimate of mucositis incidence.

The analysis of ulceration-free survival during the chemotherapy treatment is based on self-reported adherence to hygiene recommendations, which might entail an underestimation of the effect size. In addition, the study enrolled patients with a wide range of cancer diseases and general medical conditions on the one hand and various treatments on the other hand, which resulted in a wide range of degrees of effects on oral health status. In spite of all these confounding factors, hygiene proved to be a significant independent protective factor for oral health during chemotherapy.

Our analysis included a wide range of treatments (comprising various chemotherapeutic approaches, such as adjuvant, neoadjuvant, or palliative treatments) and cancer sites, with no restrictions concerning tumor recurrence, genetic profiles, age, sex, or even previous oral health condition; all these additional factors would contribute to the validity of the main conclusions, namely, the beneficial role of the oral hygiene protocol as recommended by the MASCC/ISOO guidelines.

An additional concern consists of the inclusion/exclusion criteria regarding pre-existing inflammation that we chose to impose in order to be sure to estimate the incidence of chemotherapy-induced OM. Although this constraint certainly limited the medical profiles of the patients included in the analysis, it ensured the validity of the incidence estimate, which was the main objective of the investigation.

Based on the current level of evidence reported in the medical literature, we analyzed a limited set of blood test results considered both meaningful for our investigation and useful for planning future studies. This is a potential limitation for two reasons: on the one hand, a limited set of blood tests concerning inflammation were reported; on the other hand, we reported results such as NLR (that had not been usually reported in studies of cancer cases other than head and neck cancer), even though they were also not significant in our study. This lack of significance might be a consequence of insufficient statistical power (due to the limited sample size) or a lack of change in NLR dynamics during chemotherapy. We have therefore avoided publication bias in our report, but it remains for future research to find an answer.

## 5. Conclusions

Our six-month longitudinal study confirmed the incidence of oral mucositis in Romanian cancer patients and would contribute to the growing body of evidence appraising the importance of adherence to oral hygiene protocol during chemotherapy in patients with various forms of cancer other than head and neck cancer. Our analysis confirmed the oral hygiene protocol as an independent protective factor and also substantiated the increased risk of oral ulceration due to non-adherence to strict oral hygiene recommendations by more than six times in patients of the same age, sex, baseline hygiene index, and treatment class.

Cancer patients have the agency to control their oral health along the difficult path of chemotherapy, regardless of their age, previous oral hygiene practices, or particular treatment.

## Figures and Tables

**Figure 1 jcm-13-05723-f001:**
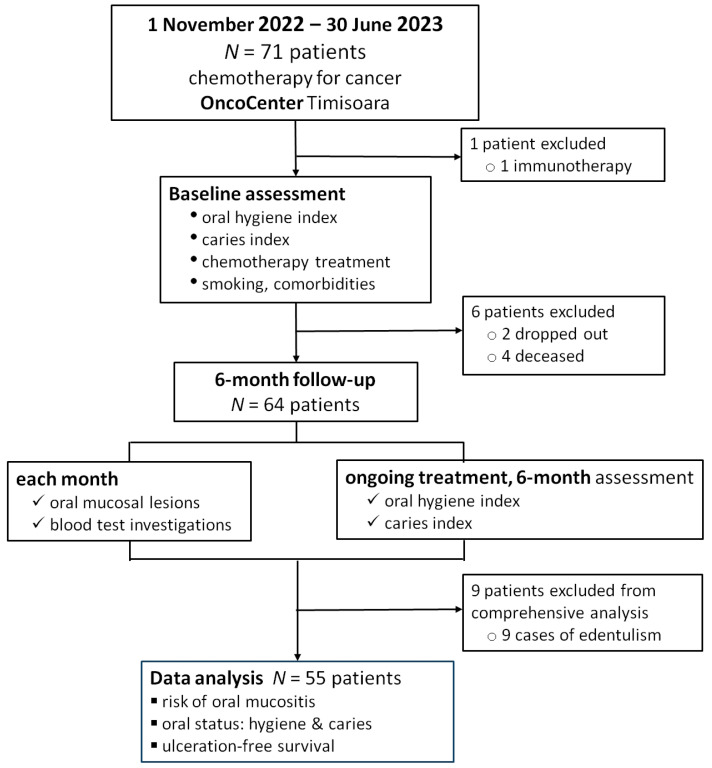
Study flowchart.

**Figure 2 jcm-13-05723-f002:**
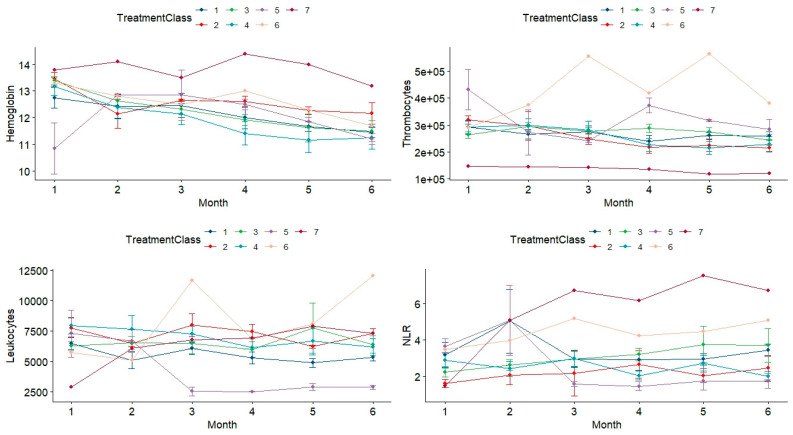
Results of monthly laboratory tests during chemotherapy treatment. High variability is apparent across the seven treatment classes.

**Figure 3 jcm-13-05723-f003:**
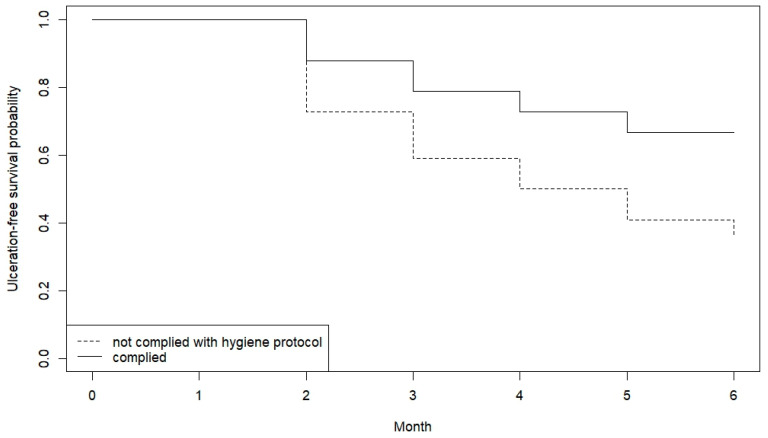
Kaplan–Meier survival time for the 55 non-edentulous patients with cancer (other than head and neck cancer) during the first 6 months of chemotherapy. The event was the first episode of oral mucositis grade 2 or higher.

**Table 1 jcm-13-05723-t001:** The dental hygiene index from Ref. [46].

Plaque Index	Criteria
0	no plaque in the cervical area of the tooth
1	plaque adhered to the cervical area adjacent to the gingival margin, recognized only by running a probe over the tooth surface
2	moderate accumulation of soft residues inside the gingival sulcus, and on the cervical area, visible with the naked eye
3	abundance of soft matter inside the gingival pocket and on the cervical area

**Table 2 jcm-13-05723-t002:** General characteristics of the cancer patients enrolled.

Characteristic/Variable		Descriptive Statistics (N = 64 Patients in Total)
Sex ^(a)^	F	48 (75%)
	M	16 (25%)
Age in years ^(b)^		59.28 ± 10.84
Living in area ^(a)^	Rural	13 (20.31%)
	Urban	51 (79.69%)
Complied with the hygiene protocol ^(a)^		39 (60.94%)
Smoking ^(a)^		23 (35.94%)
Toothlessness/edentulism ^(a)^		9 (14.06%)
Reported oral mucosal lesions ^(a),#^		34 (53.13%)
Tumor location	Breast cancer	23
	Broncho-pulmonary cancer	12
	Colorectal	10
	Uterine cancer	10
	Pancreatic cancer	2
	Prostate cancer	2
	Reno-urinary cancer	2
	Sarcoma	2
	Gastric cancer	1
Recurrent tumor ^(a)^		26 (40.63%)
Newly diagnosed tumor ^(a)^		38 (59.37%)
	Adjuvant chemotherapy	16
	Neoadjuvant chemotherapy	14
	Palliative chemotherapy	8

^(a)^ Observed frequency (percentage); ^(b)^ mean ± standard deviation; ^#^ some patients experienced multiple lesions at a time and more than one type of lesion.

**Table 3 jcm-13-05723-t003:** The worst lesion experienced.

Mucositis Grade WHO Classification [44]	Cases of Mucositis ^#^ N = 64 Patients in Total
0 (none)	30
1 (mild)	6
2 (moderate)	28
3 (severe)	0

^#^ Some patients experienced multiple lesions at a time and more than one type of lesion.

**Table 4 jcm-13-05723-t004:** Coding of treatments the participants underwent.

Code	Class of Treatment
1	Taxane mitotic inhibitors
2	Monoclonal antibodies
3	Anthracyclines-Antibiotics
4	Antimetabolites
5	Protein kinase inhibitors
6	Selective mTOR inhibitors
7	Alkylating agents

**Table 5 jcm-13-05723-t005:** Laboratory test results during the six-month follow-up (N = 64 patients in total).

	Blood Test			
	Hemoglobin ^(a)^	Leukocytes ^(a)^	Thrombocytes ^(a)^	NLR ^(a)^
Month 1	13.10 ± 1.32	6926.41 ± 2510.28	284,973.44 ± 92,719.45	2.72 ± 1.86
Month 2	12.52 ± 1.48	6614.49 ± 3792.06	288,952.38 ± 90,300.11	3.17 ± 2.29
Month 3	12.33 ± 1.50	6629.37 ± 3078.35	278,442.67 ± 113,581.23	2.91 ± 1.82
Month 4	11.86 ± 1.55	5886.38 ± 3016.51	255,515.63 ± 96,240.60	2.72 ± 1.66
Month 5	11.57 ± 1.55	5850.30 ± 3835.29	248,343.75 ± 95,526.65	2.95 ± 1.89
Month 6	11.44 ± 1.43	5881.72 ± 2539.56	244,873.44 ± 102,850.65	2.92 ± 1.77

^(a)^ Mean ± standard deviation. Abbreviation: NLR, neutrophil-to-lymphocyte ratio.

**Table 6 jcm-13-05723-t006:** Repeated measures ANOVA for laboratory test results during the six-month follow-up: month and treatment class are treated as factors while controlling for sex (N = 64 patients in total).

Blood Test	Model Significance	Month Significance	Class of Treatment Significance
Hemoglobin	<0.001 **	<0.001 **	0.002 **
Leukocytes	0.001 **	0.307	0.012 *
Thrombocytes	<0.001 **	0.036 *	<0.001 **
NLR	0.005 **	0.816	<0.001 **

Statistical significance: *, *p* < 0.05; **, *p* < 0.001.

**Table 7 jcm-13-05723-t007:** Six-month cumulative incidence rate of chemotherapy-induced mucositis among patients with cancer (other than head and neck cancer).

Event	6-Month Cumulative Incidence Rate (95%CI)
Any type of oral mucosal lesion	53.13% (40.33–65.55%)
Mucositis grade 2 or higher (painful inflammation and ulceration)	43.75% (31.58–56.67%)

Abbreviation: CI, confidence interval.

**Table 8 jcm-13-05723-t008:** Assessment of baseline characteristics and six-month changes. This assessment was performed only for patients who were not edentulous (i.e., nine edentulous patients were excluded from this analysis). DFMT stands for Decayed, Missing, and Filled Permanent Teeth.

Variable	All Patients	Complied with Oral Hygiene Protocol		*p*-Value ^(a),(b)^
	N = 55	Yes (N = 33)	No (N = 22)	
Baseline hygiene index ^(a)^	0.97 ± 0.81	0.60 ± 0.43	1.53 ± 0.93	<0.001 **
6-month hygiene index ^(a)^	1.10 ± 0.88	0.69 ± 0.48	1.72 ± 0.98	<0.001 **
Change in hygiene index ^(a)^	0.13 ± 0.12	0.09 ± 0.07	0.12 ± 0.15	0.003 **
Baseline DFMT index ^(a)^	18.65 ± 7.67	15.33 ± 6.33	23.64 ± 6.86	<0.001 **
6-month DFMT index ^(a)^	19.15 ± 7.50	15.94 ± 6.14	23.95 ± 6.85	<0.001 **
Change in DFMT index ^(a)^	0.49 ±0.81	0.61 ± 0.93	0.32 ± 0.57	0.340
Mucositis grade 2 or more ^(b)^	25 (45.5%)	11 (33.3%)	14 (63.5%)	0.027 *

^(a)^ Mean ± standard deviation; non-normal distribution; Mann–Whitney U test; ^(b)^ observed frequency (percentage); asymptotic chi-square test; statistical significance: *, *p* < 0.05; **, *p* < 0.001.

**Table 9 jcm-13-05723-t009:** Ulceration-free survival analysis for the 55 non-edentulous patients with cancer (other than head and neck cancer) during the first 6 months of chemotherapy.

Compliance with Hygiene Protocol	Ulceration–Free Mean Survival Time in Months		Ulceration–Free Median Survival Time in Months	
	Estimate ± Std. Err.	95%CI	Estimate ± Std. Err.	95%CI
No	4.227 ± 0.377	(3.488–4.966)	4.0 ± 1.173	(1.702–6.298)
Yes	5.061 ± 0.223	(4.289–5.165)	–	–
	Log-rank test for the incidence rates: chi-square = 4.943 (1 df), *p* = 0.026 *			

Abbreviations: CI, confidence interval; df, degrees of freedom; Std. err., standard error; statistical significance: *, *p* < 0.05.

**Table 10 jcm-13-05723-t010:** Cox regression analysis of ulceration-free survival during the first 6 months of chemotherapy in patients with cancer (other than head and neck cancer), with adherence to the hygiene protocol as an independent predictor.

**Cox proportional-hazards model**: first ulceration month~compliance with hygiene protocol				
Controlling for: age, sex, baseline hygiene index, class of treatment				
	**Predictor**	**B ± Std. Err**	** *p* ** **-Value**	**HR (95%CI)**
	Compliance with hygiene protocol	−1.874 ± 0.584	0.00134 **	0.154 (0.049–0.483)

Abbreviations: B ± Std. err, coefficient of regression ± standard error; CI, confidence interval; HR, hazard ratio; statistical significance: ** *p* < 0.01.

## Data Availability

The data presented in this study are available on request from the first author due to privacy restrictions.

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
