# Peer review of "Oral Mucositis in Adult Cancer Patients Undergoing Chemotherapy: Six-Month On-Treatment Follow-Up"

_jcm, 2024, doi:10.3390/jcm13195723_

Round 1

Reviewer 1 Report

Comments and Suggestions for Authors

The article deals with a classic hypothesis in the management and prevention of the appearance of oral mucositis in oncologic patients.

Some decades ago, at the beginning of the century, articles were frequently published, highlighting the importance of dental care in these patients to avoid the appearance of oral mucositis and limit its severity. In recent years, different approaches and treatments have been investigated to prevent the appearance of oral mucositis with more advanced therapies not necessarily linked to oral hygiene, such as laser, taking for granted that oral care should be incorporated into every oncological protocol and treatment.

Nevertheless, although the article's hypothesis has been tested in multiple previous experiences, it is well developed, reads smoothly, and provides interesting data that can remind the scientific community not to forget the importance of oral health and adherence to hygiene protocols in this type of patient.

Author Response

Dear Reviewer,

Thank you for your appreciation of our study. We also thank you for the time and effort spent on helping us to prepare a new version of the manuscript, which we believe is more appropriate for publication.

To emphasize the importance of educating patients to follow hygiene protocols recommended by professional guidelines, we added a paragraph to the final conclusions:

"Cancer patients have the agency to control their oral health along the difficult path of chemotherapy, regardless of their age, previous oral hygiene practices, or particular treatment."

Please also find the revised manuscript with highlighted changes.

The Authors

Reviewer 2 Report

Comments and Suggestions for Authors

Congratulations to the authors for the interesting study they performed.

1. Please follow report guidelines based on your study design and outcome. STROBE? REMARK? I'm not sure which one is more appropriate, I think STROBE as I don't think this is prospective study in which there s an allocation

2. Consider cite and discuss your results compared to https://pubmed.ncbi.nlm.nih.gov/38923624/

3. "Three-way analysis of variance (ANOVA) was applied to analyze the repeated 182 measures of blood tests and assess the statistical significance of observed trend differ- 183 ences between the classes of treatment." Do you mean Repeated measure anova? Which are the classes of treatments?

4. Change gender to sex

5. Use C-index together with AIC

6. What is the difference between aphthous stomatitis and mucositis?

7. RMA requires some assumptions, were these checked? Table 5 and 6 should be merged to make a more informative table, faster to understand

8. The class of treatment significance is difficult to understand

9. Univariate and multivariate analysis show indicate which variables were included in the model and showing all the variables results. Authors should run first univariate analysis for the all variables and those p-value<0.100 should be included into the cox regression. 

10. AIC is in method but not in results

Comments on the Quality of English Language

Minor

Author Response

Thank you for your review. Please find the revised manuscript and the answers to your concerns, attached herewith.

The Authors

Reviewer 3 Report

Comments and Suggestions for Authors

Review of Oral Mucositis in Adult Cancer Patients Undergoing Chemotherapy: Six-Month On-Treatment Follow-Up

Line 20: please explain to the readers why “other than head and neck cancer”, since this is the topic of main interest where the effects of chemoradiation therapy are stronger 

Line 42: review the punctuation

Line 43-47: too wordy, please rephrase

Line 50-78: please shorten up and synthesize this paragraph since it is too wordy for the introduction subheading and the aim of the study is not related with the biochemical pathways of inflammation and epithelial damage

Line 64: please add the estimated overall incidence and relevant literature

Line 67: “Genetic determinants and oral microflora also play a role in the pathogeny of mucositis”: please expand

Line 68-70: “Inflammation markers such as the neutrophil-to-lymphocyte ratio (NLR) have been shown to be predictive factors for radiotherapy-induced OM in head and neck cancer patients”: not so relevant for this study if you exclude HN cancers

Line 79: “pathogenicity” is incorrect in this context

Line 84: replace “MASCC/ISOO guidelines” with “Multinational Association of Supportive Care in Cancer (MASCC) / International Society of Oral Oncology (ISOO)”

Line 86: “The Romanian population has particular characteristics in terms of oral health, knowledge of oral care, and cultural backgrounds” please explain 

Line 105: please better specify the follow-up and the setting. I mean, it can be an adjuvant chemotherapy after surgical o radiation therapy, or exclusive chemotherapy as a first line or neoadjuvant. Please specify. 

Line 124-126: baseline assessment includes smoking and comorbidities evaluation as per study flow-chart but there is no mention of it in this paragraph.

Line 203: what types of neoplasms were the patients affected by? Please specify

Line 342: what were the possible reasons for non-compliance? 

Comment:

Dear Editors,

Thank you for the opportunity to review this paper. 

This is a prospective longitudinal six-month study to estimate the cumulative incidence of oral mucositis in 55 adult patients with cancer other than head and neck receiving chemotherapy. The authors concluded that adherence to the oral hygiene protocol was an independent protective factor against oral ulceration, suggesting compliance with the hygiene recommendations would decrease the OM risk by more than six times, compared to non-compliance, adjusted for age, sex, baseline hygiene index, and treatment class. The topic is of interest as educational guidelines represent an effective intervention to reduce OM severity and improve the quality of life of oncology patients receiving chemotherapy. 

Articles with related topic:

-              Pilas SA, Kurt S. Evaluation of oral mucositis level and affecting factors in cancer patients receiving chemotherapy. Support Care Cancer. 2024 Aug 20;32(9):597. doi: 10.1007/s00520-024-08812-9. PMID: 39162830.

-              Elsehrawy MG, Ibrahim NM, Eltahry SI, Elgazzar SE. Impact of Educational Guidelines on Oral Mucositis Severity and Quality of Life in Oncology Patients Receiving Chemotherapy: A Quasi-Experimental Study. Asian Pac J Cancer Prev. 2024 Jul 1;25(7):2427-2438. doi: 10.31557/APJCP.2024.25.7.2427. PMID: 39068577.

Self-reference from the authors:

No self-reference. 

Aim of the paper: 

Estimate the cumulative incidence of OM during the first six months of chemotherapy in adult patients with cancer other than head and neck cancer and assess changes in oral health during chemotherapy. 

Strengths of the study: 

-              Adult population (most of the study includes pediatric population)

-              Uniform management of patients at the same institution with the same clinical and laboratory tests during the six months of follow-up

Major concerns: 

-              Small sample (55 patients)

-              Heterogeneous sample: too many chemotherapies classes 

Comments on the Quality of English Language

English is fine, minor editing is needed 

Author Response

(The authors gave the same response as above.)

Reviewer 4 Report

Comments and Suggestions for Authors

The research is very interesting, and the conclusions seem relevant, so I would like to make sure they are based on correct analysis and interpretation of the results.

The manuscript lacks data on the type of cancers patients suffered from. Also missing is a univariate analysis of how age, gender, and type of therapy used affected the oral cavity. Compliance with hygiene protocol was not fully defined. Did it mean that the patient rinsed his mouth twice a day and additionally brushed his teeth twice a day at least five days a week? What about flossing? Was denture hygiene taken into account in edentulous patients? What about treatment of mucositis? Did all patients follow doctor’s recommendations? Were similar lesions treated in a similar way? The table with types of mucositis is unclear. The authors list various mucosal lesions there, with one subgroup called mucositis, but all groups were classified as oral mucositis? Do we know what types of lesions appeared in each patient? I guess that oral hygiene protocol may have limited efficacy in the prevention of angular cheilitis.

The authors assigned codes (1-7) to the different chemotherapy regimens, but they do not clarify whether these codes represent a ranking (e.g., 1 being the lowest risk and 7 the highest) or if they are simply categorical identifiers.

The authors state that the Cox model was adjusted for certain variables, but they do not clarify how these variables were selected or if any multicollinearity issues were checked, moreover they do not specify the individual contributions of these variables to the risk of mucositis.

Age and compliance with the oral hygiene protocol could be correlated. Older patients might have different levels of adherence due to factors like dexterity, cognitive function, or the severity of chemotherapy side effects.

Stronger chemotherapy agents may lead to more severe side effects, potentially reducing patients' ability to adhere to oral hygiene protocols. If both chemotherapy type and adherence were included in the model, there’s a risk of multicollinearity.

If these issues were not adequately addressed in the study, the conclusions drawn from the Cox model could be misleading. It would be helpful if the full study provided more details on the aspects mentioned above.

Author Response

(The authors gave the same response as above.)

Reviewer 5 Report

Comments and Suggestions for Authors

The manuscript presents a valuable study on the incidence and management of oral mucositis (OM) in adult cancer patients undergoing chemotherapy. However, it requires major revisions to improve the clarity and accuracy of the content.

Specific Comments:

Line 24: The phrase “Sixty-four patients participated.” should be moved to the methods subsection.

Line 36-37: Replace the keywords “basic oral care,” “cancer,” and “patient education” with more specific terms.

Line 43: The references need to be corrected. For such a general sentence, the number of references is excessive.

Line 102: It is essential to explicitly state the type of study (e.g., cohort study, case-control study...).

Lines 116-117: While using normal CRP levels (<5 mg/L) as an inclusion criterion is a reasonable approach to minimize confounding from pre-existing inflammation, it raises concerns regarding the exclusion of patients with elevated CRP due to cancer itself. The authors should clarify how they distinguished between CRP elevation due to cancer versus other causes. Additionally, the choice of CRP as the sole marker of inflammation should be justified. Discuss why other markers, such as ESR, were not included, and consider whether including multiple markers might have provided a more comprehensive assessment.

Lines 156-157: The criteria for compliance with the oral hygiene protocol need clarification. The objective basis for these criteria should be discussed to enhance the study's reproducibility and validity.

Lines 297-301: The sentence inaccurately compares the incidence of chemotherapy-induced oral mucositis from your study with the prevalence of OM from a meta-analysis. Incidence and prevalence are distinct epidemiological measures. Please revise this section to accurately reflect the differences and ensure that comparisons are appropriately made.

Author Response

(The authors gave the same response as above.)

Reviewer 6 Report

Comments and Suggestions for Authors

Review Report

 Manuscript ID jcm-318

 Haut du formulaire

Journal

JCM (ISSN 2077-0383)  4868

Type Article

Title Oral Mucositis in Adult Cancer Patients Undergoing Chemotherapy: Six-Month On-Treatment Follow-Up

Authors Adriana Padure , Raluca Horhat * , Ioana Cristina Talpos-Niculescu , Roxana Scheusan , Mirella D. Anghel , Laura Cristina Rusu , Diana Lungeanu

Section Dentistry, Oral Surgery and Oral Medicine

Bas du formulaire

This article proposes a six-month longitudinal study to estimate the cumulative incidence of  OM following chemotherapy in 64 adult patients with cancer other than head and  neck cancer. The secondary objectives are: firstly to examine the oral health status of these patients and their  evolution during chemotherapy, thanks to oral health indices, secondly appreciate respect for  oral hygiene protocol prescribed during chemotherapy and finally thirdly monitor the evolution of ulcerations in these  patients.

The main criticism of this article is that the study includes heterogeneities on various parameters, such as   chemotherapy regimens, administration of antimicrobials during chemotherapy. Concerning patients, more information on their general conditions and locally the periodontal state of the oral cavity is also lacking. It is desirable   to implement standardized protocols in this type of study to achieve greater consistency and uniformity in research results.

Another parameter concerning this type of study limits the scope of the results. These are the

specific prevention/treatment protocols. A bias lies in the more or less variable and verifiable follow-up of these protocols by different patients. Furthermore, to date, there is no obvious correlation between oral hygiene and prevention and/or treatment of mucositis. For these various reasons, reservations are necessary to conclude the article. However, beyond these limitations, this study can contribute to a better understanding of the role of the oral microbiota in the appearance and progression of mucositis.

Author Response

(The authors gave the same response as above.)

Round 2

Reviewer 2 Report

Comments and Suggestions for Authors

Authors completed my comments. 

Please include whether RMAnova was checked for sphericity

Reviewer 3 Report

Comments and Suggestions for Authors

Thanks for your corrections, I think that now the paper has significantly improved. 

Comments on the Quality of English Language

Minor editing 

Reviewer 5 Report

Comments and Suggestions for Authors

All relevant amendments have been performed.

Reviewer 6 Report

Comments and Suggestions for Authors the article thus worded presents the qualities necessary to interest readers.